# Flexible simultaneous mesoscale two-photon imaging of neural activity at high speeds

Mitchell Clough[1], Ichun Anderson Chen[1,2], Seong-Wook Park[1,2], Allison M. Ahrens[3], Jeffrey N. Stirman[4], Spencer L. Smith[5] & Jerry L. Chen[1,2,3✉]

Understanding brain function requires monitoring local and global brain dynamics. Two-photon imaging of the brain across mesoscopic scales has presented trade-offs between imaging area and acquisition speed. We describe a flexible cellular resolution two-photon microscope capable of simultaneous video rate acquisition of four independently targetable brain regions spanning an approximate five-millimeter field of view. With this system, we demonstrate the ability to measure calcium activity across mouse sensorimotor cortex at behaviorally relevant timescales.

[1] Department of Biomedical Engineering, Boston University, Boston, MA 02215, USA. [2] Center for Neurophotonics, Boston University, Boston, MA 02215, USA. [3] Department of Biology, Boston University, Boston, MA 02215, USA. [4] Neuroscience Center, University of North Carolina School of Medicine, Chapel Hill, NC 27599, USA. [5] Department of Electrical and Computer Engineering, University of California Santa Barbara, Santa Barbara, CA 93106, USA. ✉email: jerry@chen-lab.org

In order to understand how the brain functions as a whole to process information and carry out behavior, it is necessary to monitor the patterns of neuronal activity across the brain with cellular resolution. Two-photon microscopy with calcium indicators has become a standard tool in neuroscience for functional in vivo imaging of neuronal populations in awake behaving animals[1]. Recently, the development of new two-photon microscopes has enabled imaging of increasingly large numbers of neurons across different brain areas. This has been achieved through the design and fabrication of custom optical elements that support imaging over several millimeter fields of view while maintaining cellular resolution[2,3]. However, the current use of scanning strategies with single foci excitation produces a trade-off between the number of imaging areas and the overall rate of acquisition[3,4]. Total frame rates of up to ~10 Hz have typically been achieved, which can limit the types of neuronal dynamics that can be investigated. Sensorimotor behaviors such as whisking, sniffing, eye movements, and locomotion have been linked to ongoing neuronal activity that fires on the order of 4–12 Hz[5]. Deconvolution of calcium transients can improve the temporal precision of spiking events to within <100 ms[6]. Therefore, to link these estimated spikes to behavior, sampling rates need to be at a sufficient Nyquist frequency (>30 Hz). Parallel scanning using multi-foci excitation can achieve true simultaneous multi-area imaging at high scanning speeds[7,8]. So far, large field of view (FOV) multi-foci two-photon microscopes have been designed with fixed beam configurations to match spatially arranged detection schemes[9]. This restricts a user's ability to target specific neuronal populations of interest across the FOV and limits the ability to resolve fluorescence at increasing depths due to spatial cross talk from light scatter.

To address the gap in these technologies, we designed and constructed a quad-area large FOV two-photon microscope (Quadroscope) to achieve simultaneous, video-rate cellular-resolution imaging across four independently assignable FOVs. We demonstrate its utility in carrying out population imaging across mouse sensorimotor cortex during whisker-guided behavior at behaviorally relevant timescales.

## Results

**Large field of view four area two-photon microscope design.** The Quadroscope employs a custom optical design to achieve a 0.52 numerical aperture (NA) across a 4.8 mm FOV, sufficient to span mouse sensorimotor cortex (Fig. 1, Supplementary Figs. 1–2). Diffraction-limited performance was optimized across an excitation wavelength range of $\lambda = 920$–1040 nm for two-photon imaging of green, yellow, and red fluorescent molecules. For scanning and independent positioning of multiple beams across all three axes, we employed a combination of two strategies. First, two independent scanning arms were deployed. Each scanning arm consisted of a resonant scanner for fast x-scanning followed by a 2D steering mirror. The 2D steering mirror provides both slow y-scanning as well as positioning across the entire FOV through fixed angular offsets along the optical axis of the system. The two scanning arms are combined prior to the tube lens using a polarizing beam splitter. Through this, each scanning arm can function independently and can also be synchronized by locking the two resonant scanners in phase. The use of two independent scanning arms means that the imaging areas from each arm can be positioned anywhere within the total FOV such that imaging areas from the first arm can also overlap with imaging areas from the second arm.

Positionable, non-overlapping imaging areas can further be deployed by inserting movable coupling units, named 'focal plane units' (FPUs), into each scanning arm (Fig. 1b). Independent focusing is achieved with electrically tunable lenses (ETLs) installed into each of the FPUs, which provides a z-range of up to 480 μm. Lateral positioning of each beam is achieved through x/y-translation stages that introduce an offset of the respective beam from the optical axis of the first scan lens. This converts the offset into a pivoting angle of the beam around the resonant scanning mirror, which is further translated into lateral movement of the corresponding imaging sub-area below the objective. For each FPU-assigned beam belonging to one scan arm, polarization is then matched in order for those beams to be combined with those belonging to the other scan arm. Adjustment of the polarization for each FPU-assigned beam independently controls excitation power of that beam. In the Quadroscope, two FPUs were installed on each scan arm to provide four independent scanning regions across a 4.8 mm total FOV. Within each scan arm, the scan field of the first scan lens defines the maximum separation between each FPU-beam (3 mm) while the scan angle of the resonance scanner defines the imaging field for each FPU-area (0.75 mm) (Fig. 1c). With these specifications, the system is designed to achieve simultaneous video rate imaging of four regions totaling $1.5 \times 1.5$ mm$^2$ within the entire 4.8 mm field of view of the system (Fig. 1d).

To achieve simultaneous imaging across four FOVs with minimal cross talk and maximal flexibility in beam positioning, multiple foci were generated using spatiotemporal multiplexing (Fig. 1e). Emission fluorescence was collected using a single hybrid photomultiplier tube and signals were de-multiplexed using a field-programmable gate array. In order to resolve the 2–3 ns fluorescence lifetimes of existing genetically encoded calcium indicators (GECIs), we used custom Ytterbium (Yb)-fiber lasers, each with a 31.25 MHz repetition rate to provide an 8 ns inter-pulse interval between each of the four imaging areas with minimal (<3%) cross talk (Supplementary Fig. 3). One laser is tuned to 920 nm to provide two-photon excitation of green GECIs[10]. The other laser is tuned to 1040 nm to provide two-photon excitation of new classes of red and yellow GECIs, including RCaMP1.07 and jYCaMP1s[11,12].

**Characterizing optical performance of the Quadroscope.** We first measured the optical performance of the system. While the optical path of the system is designed for 10 mm scanning mirrors with a clear aperture of 7.2 mm, we used commercially available 9 mm 6 khz resonance scanners with a clear aperture of 6.4 mm to maximize scan speeds. The smaller clear aperture produced a beam diameter at the back pupil plane of the objective that is 12% under-filled, resulting in an effective excitation NA of 0.46. As expected from the decrease in excitation NA, we were able to achieve lateral and axial resolutions of $0.91 \pm 0.02$ μm ($n = 8$) and $10.51 \pm 0.43$ μm ($n = 8$) respectively at the center of the FOV and $1.22 \pm 0.03$ μm ($n = 32$) and $12.46 \pm 0.26$ μm ($n = 32$) respectively at the edges of the FOV (Supplementary Fig. 4). Due to the natural curvature of the mouse cortex, the field curvature of the system was not optimized (Supplementary Fig. 5). Instead, independent ETL focusing was used to account for differences in focal depth between FPU-areas. Excitation performance at different remote focusing depths was comparable when imaging at the center and edge of the FOV, as predicted from simulations (Supplementary Fig. 6). The effective detection NA varied as a function of distance from the center of the FOV (Supplementary Fig. 7). Overall, the optical performance of the system across the imaging FOV is suitable for cellular resolution imaging.

**Simultaneous imaging of four areas during task behavior.** To demonstrate the use of the Quadroscope for in vivo imaging of

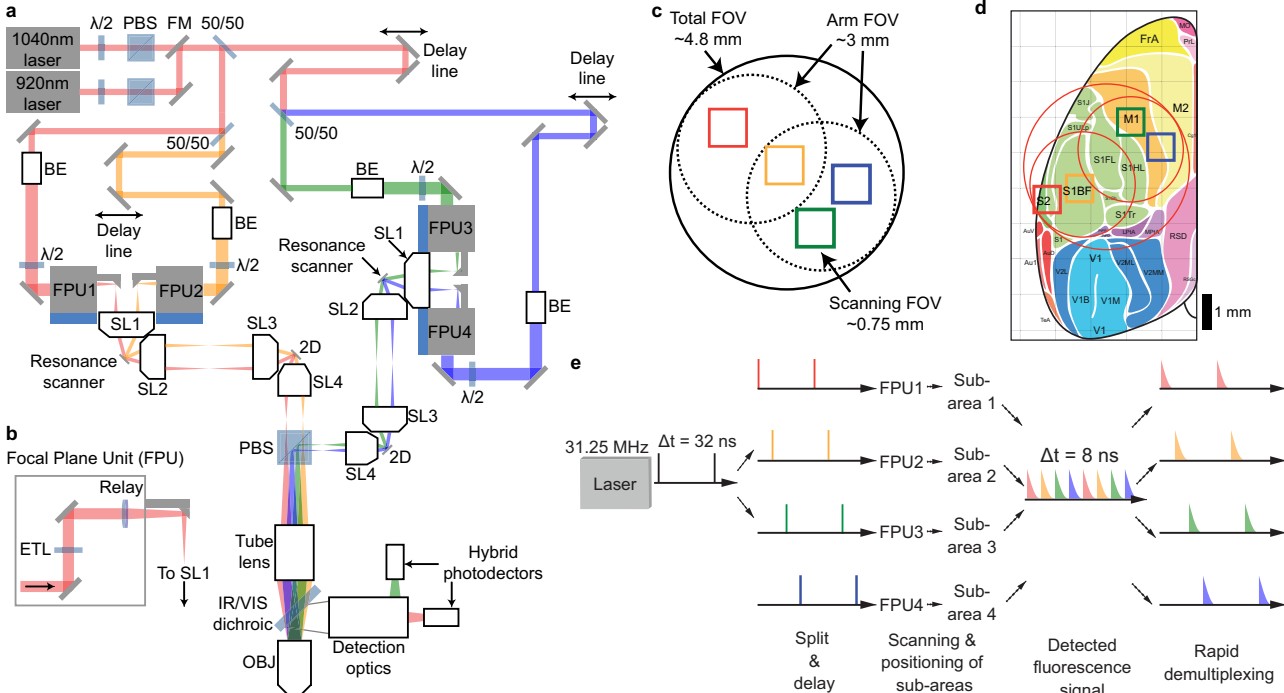

**Fig. 1 Large field of view quad area two-photon microscope system. a** Schematic of the Quadroscope system. Laser source is selected by a flip mounted mirror (FM). Laser output is then split into four beam paths (beam 1—red, 2—orange, 3—green, 4—blue) using 50/50 beam splitters (50/50). Each beam is delayed 8 ns relative to the other. Each beam is magnified by a beam expander (BE) entering into the focal plane units (FPUs). Half-wave plates ($\lambda/2$) are used in conjunction with a polarizing beam splitter (PBS) to control power of each beam. Two scan engines each control two of the imaging regions. Each scan engine consists of three commercial scan lenses (SL1-3), a resonance scanner, a 2D scanning mirror, and a custom scan lens (SL4) before being combined using the PBS onto a custom tube lens. The beams enter a custom objective (OBJ) before arriving at the sample. A long-pass dichroic (IR/VIS Dichroic) is used to separate the excitation and emission light. **b** FPU design. Each FPU is mounted on a motorized XY stage and contains an electrically tunable lens (ETL) and relay lens. **c** Schematic of scan area positioning. Scanning regions of up to ~0.75 mm$^2$ can be achieved for each of the four sub-areas. Each scan arm has a 3 mm arm field of view (FOV) in which sub-areas can be repositioned using the FPU. The arm FOV can be positioned anywhere within the 4.8 mm total FOV using 2D scanning mirrors. **d** Scanning areas and FOVs overlaid across an atlas of mouse cortex, adapted from ref. [25] with permission. **e** Spatiotemporal multiplexing of four beams. A 31.25 MHz laser source is split into four beams and delayed to achieve 8 ns inter-pulse intervals. The detected fluorescence signal is then de-multiplexed and assigned to each imaging area.

neuronal population activity, we performed high speed simultaneous calcium imaging across targeted areas in mouse sensorimotor cortex (Fig. 2, Supplementary Fig. 8). We virally expressed jYCaMP1s or GCaMP7f in primary somatosensory cortex (S1), secondary somatosensory cortex (S2), and primary motor cortex (M1). A curved cover glass window ('crystal skull') was implanted to provide optical access to a complete hemisphere of mouse dorsal cortex[13]. Through a combination of FPU translation, 2D steering mirror angular offset, and ETL focusing, we selectively targeted each expression area. Individual neurons were clearly identifiable and distinguishable across all sub-areas. By performing simultaneous imaging across four areas at 30 Hz, we were able to observe calcium transients from 150 to 200 neurons in each area in the awake, behaving animal.

For each sub-area, we applied an average excitation power of 50–75 mW, similar to standard 2P microscope systems using 80 Mhz Ti:sapphire lasers. While the lower repetition rate lasers results in a 2.5-fold lower pulse rate per pixel dwell time (~5 pulses per 0.16 us) compared to Ti:sapphire lasers, this is compensated by a 2.5-fold higher peak power per pulse (1.6–2.4 nJ) providing comparable excitation during high speed imaging. Imaging at 50–75 mW per sub-area results in ~200–300 mW of total power into the brain. Sustained imaging using a single 400 mW foci within one imaging region has been shown to induce lasting photo-damage[14]. We checked if damage could be induced with Quadroscope imaging under normal operation if equivalent total power was instead divided between four beams distributed across

the cortex. Continuous, simultaneous four-beam imaging was carried out at 75 mW per beam for ~30 min with beams distributed across S1, S2, and M1 (Supplementary Fig. 9). Brains were extracted 16 h after imaging, and immunohistochemistry was performed for astrocytic (anti-GFAP), microglial (anti-Iba1), heat shock (anti HSP-70/72), and apoptotic pathway (anti-Caspase-3) activation. No significant difference between stimulated and neighboring control areas was observed for each of the markers. This demonstrates that sustained, distributed multi-region imaging does not induce significant tissue damage and is safe for long-term imaging experiments.

To demonstrate novel biological applications of the Quadroscope, we tracked neuronal activity during sensorimotor behavior (Fig. 3). Mice were trained to perform a whisker-based texture discrimination task where mice rhythmically whisk to discriminate the roughness or smoothness of different sandpaper textures[7,15]. Using high-speed videography, we monitored changes in whisker angle related to rhythmic 7–12 Hz anticipatory whisking which we decomposed into specific components reflecting the phase, amplitude, and variations in the midpoint of the whisker cycle by Hilbert transform. Neuronal activity in S1 has been previously shown to be locked to specific phases of the whisk cycle while neurons in M1 track the amplitude and midpoint of whisking[16,17]. Analysis of deconvolved calcium activity of simultaneously imaged neurons across S1 and M1 confirmed that imaging rates of 30 Hz were sufficient to distinguish between neurons with preferences for specific

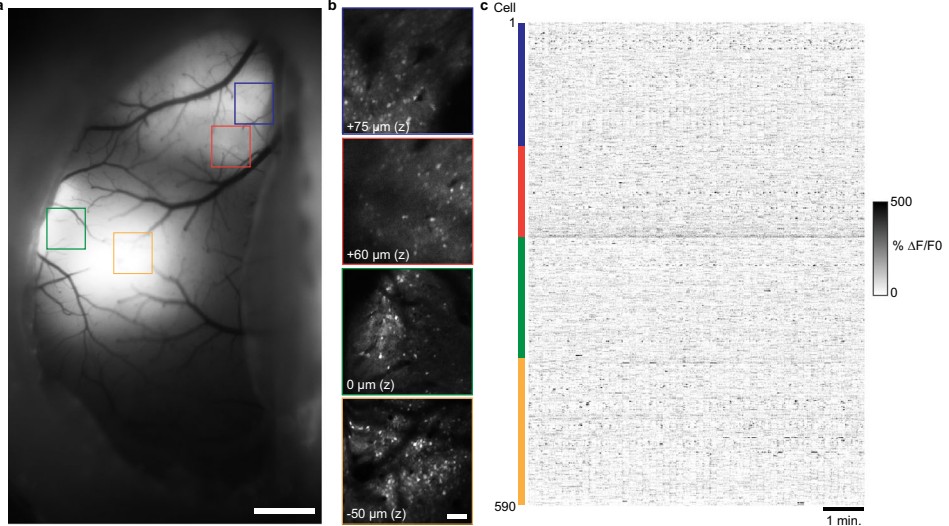

**Fig. 2 Quad area two-photon calcium imaging of jYCaMP1s in sensorimotor cortex in the awake mouse using a 1040 nm laser source. a** Wide-field fluorescence image of mouse sensorimotor cortex showing jYCamp1s expression through an implanted crystal skull. **b** Two-photon images of the regions expressing jYCamp1s displayed in (**a**) acquired with the Quadroscope. Imaging depth reflects relative focal plane positioned with each electrically tunable lens at the imaging region. Images in (**a**) and (**b**) have been obtained from five animals with similar results. **c** Calcium transients in the awake animal from cortical neurons imaged across the four areas in (**b**), simultaneously acquired at 30 Hz. Scale bars: 1 mm (**a**), 100 μm (**b**).

whisking-related components. This demonstrates the Quadroscope is capable of tracking neuronal responses across distant cortical areas at behaviorally relevant timescales.

## Discussion

In summary, we demonstrate that the Quadroscope is a powerful new system to simultaneously capture neuronal dynamics across multiple cortical areas with high temporal resolution. This system is a member of a new suite of mesoscopic imaging systems that provides simultaneous imaging across large FOVs using dual independent scan engines[18]. The Dual Independent Enhanced Scan Engines, Large-Field Two-Photon (Diesel2p) mesoscope uses a similar objective and similar optical subassemblies, incorporating adaptive optics to provide improved resolution for two-area imaging. By introducing FPU-based beam combination in the Quadroscope, we readily allow for increased number of imaging FOVs, providing a scalable solution solely dictated by the repetition rate of the laser source needed for spatiotemporal demultiplexing. High power laser sources below 20 MHz combined with modular time-multiplexing approaches can be used to image 8–16 regions tiled across multiple areas or depths at 30 Hz[19]. Customized systems based on large FOV optics and dual independent scan engines demonstrate that shared design concepts can facilitate the dissemination of mesoscopes with increasing degrees of flexibility for applications across multiple experimental conditions. The ability to maintain high frame rates while imaging across distant imaging regions provides the distinct ability to track inter-area activity patterns across multiple cortical areas at timescales relevant to behavior. Such scan speeds are further necessary with the development of faster genetically encoded indicators to more precisely follow activity and neural transmission[20,21]. Overall, the versatility and modularity of this system will enable the pursuit of a range of neurobiological questions that seek to bridge local and global brain function.

## Methods

**Microscope design**. We performed optical design and evaluation using Zemax OpticStudio software (Zemax LLC) and mechanical design using Autodesk Inventor (Autodesk Inc.). An ultrashort-pulsed 31.25 MHz repetition rate, 2 W,

920 nm fiber laser (Alcor, Spark Lasers) and 7.8 W, 1040 nm fiber laser (Altair, Spark Lasers) were used as the illumination sources. The lasers' built in group delay dispersion pre-compensation was used to achieve a pulse duration of ~150 fs at the sample plane. The total illumination power delivered into the system was adjusted by rotating a half-wave plate (AHWP05M-980, Thorlabs) relative to a polarizing beam splitter cube (CCM1-PBS253/M, Thorlabs). The half-wave plate was rotated using a manual rotation mount (CRM1P/M, Thorlabs) and was rotated to allow the maximum amount of power possible into the system.

The laser beam was split into four distinct beam paths by using three ultrafast laser beam splitters (10RQ00UB.4, Newport). The first beam path was relayed directly to the rest of the microscope after splitting. For each subsequent beam an additional ~2.4 m of path length was inserted to achieve an 8 ns inter-pulse interval between each beam when combined at the objective. Each beam's diameter was adjusted by using pairs of achromatic lenses (AC254 lenses of various focal lengths, Thorlabs) to apply the necessary magnification to ensure an 8 mm beam diameter going into the FPUs. Half-wave plates (AHWP10M-980, Thorlabs) installed in motorized rotation mounts (PRM1/MZ8, Thorlabs) were positioned prior to each FPU to adjust the polarization of each beam. These half-wave plates, in combination with a polarizing beam splitter positioned prior to the tube lens, controlled the power level of each individual beam arriving at the sample. The FPU's were mounted on XY stages (8MTF-102LS05, Standa) to enable movement of the imaging region in the sample plane in *X* and *Y*. In each FPU a focus tunable lens (EL-16-40-TC) was conjugated to a resonance scanner (SC30-10×9-6-6, Electro-Optical Products Corp.) by a relay lens (AC254-100-B-ML, Thorlabs) and scan lens (S4LFT0089/094, Sill Optics GmbH). Positioning the focus tunable lenses in this way ensures that no changes in magnification occur when varying the depth of imaging. Between the relay lens and scan lens, a fold mirror (MRA10-M01, Thorlabs) was used to reflect the light vertically into the scan engine which translates FPU movement in *X* and *Y* to an angular tilt of the beam onto the resonance scanner. The resonance scanner performed fast axis scanning in the sample plane and was conjugated to a 2D beam steering mirror (OIM5010, Optics in Motion LLC) used for slow axis scanning and overall *X* and *Y* positioning of the imaging regions of interest. Conjugation between the resonance scanner and 2D mirror was achieved by a pair of scan lenses (S4LFT0089/094, Sill Optics GmbH). The 2D mirror was conjugated to the back pupil plane of the objective by a custom designed scan lens and tube lens (Supplementary Fig. 2). The custom scan lens and tube lens were designed to achieve diffraction limited RMS wavefront error across the entire 4.8 mm FOV using any combination of FPU and 2D mirror beam steering at 920 nm and 1040 nm when the ETL was focused at 0 μm offset. Field curvature performance was not optimized. Between the scan lens and tube lens a custom polarizing beam splitter was used to combine the beams from the two separate scan engines. The polarizing beam splitter also rejects any unwanted excess power from any of the four beams controlling the amount of excitation power reaching the sample. The custom objective[18], scan lens, polarizing beam splitter, and tube lens were all manufactured by Rocky Mountain Instrument Co. Between the tube lens and objective, a large mirror (84-438, Edmund Optics) was used to reflect light down through a long-pass dichroic (Cold Light Mirror KS 93/45°, Qioptiq) and into the objective.

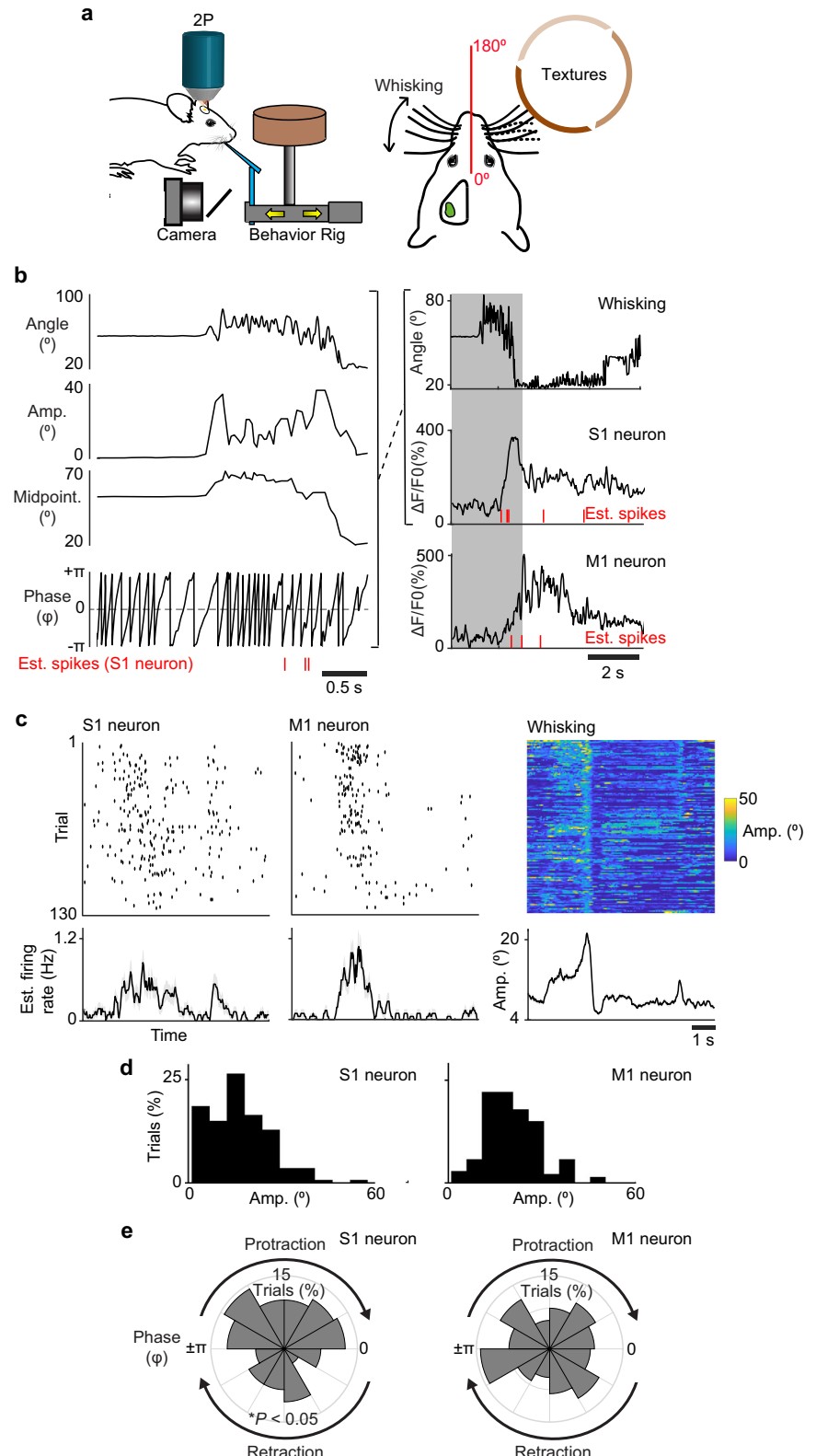

Movement of the sample relative to the microscope was achieved by mounting a lifting stage (HT160-DC, Steinmeyer Mechatronik GmbH) on top of an XY stage (KT310-DC, Steinmeyer Mechatronik GmbH) allowing for movement of the sample in three axes. The emitted fluorescence from the sample was collected by the custom objective and reflected into the detection system by the long-pass dichroic (Cold Light Mirror KS 93/45°, Qioptiq). An achromatic lens (AC508-150-A-ML, Thorlabs) was used to focus the emitted light into the detection path. A short-pass filter (FF01-720/SP-50, Semrock) rejected excitation light from the detection path and a long-pass dichroic (FF573-DI01-50×50, Semrock) split red and green fluorescent light into two separate paths. Following the dichroic, three lenses were used to focus the light onto a hybrid photo-detector with a 5 mm diameter active area (R11322U-40, Hamamatsu Photonics). The three lenses included an achromat (AC508-150-A), another achromat (AC254-030-A, Thorlabs), and an aspheric lens (ACL2018-A, Thorlabs). The aspheric lens was glued into a custom aluminum mount with external threading by using optical glue. The custom mount was connected via the internal threads of the photo-

**Fig. 3 Behavior-related responses of simultaneously imaged neurons in primary somatosensory and motor cortex. a** Schematic of Quadroscope imaging during a whisker-based texture discrimination task. Calcium imaging is performed across S1 and M1 while whisker movement is monitored using high speed videography. **b** Single trial measurement of whisker angle decomposed into the amplitude, phase, and variations in the midpoint of the whisking cycle. Raw and deconvolved calcium signals of an example S1 and M1 neuron are shown aligned to phases in the whisking cycle. **c** Estimated spikes across behavioral trials along with average estimated firing rate across the trial period are shown for example S1 and M1 neurons. Whisking amplitude across trials is also shown (right). **d** Estimated spikes binned according to whisking amplitude. M1 neuron exhibits preferred spiking to whisking amplitude. **e** Estimated spikes binned according to whisking phase. S1 neuron exhibits preferred spiking to protraction phase of the whisk cycle (*$P < 0.05$, Rayleigh test).

detector to mount the aspheric lens next to the active area of the detector for the highest collection efficiency possible.

The data were acquired from the detectors using a digital I/O board (NI6587, National Instruments) and then processed using an FPGA (PXIe-7961R, National Instruments). A custom written FPGA program was created using the LabView FPGA module (National Instruments). The digital I/O board collected the signals from the detectors and synchronization signals from both resonance scanners and the laser. The laser synchronization signal was delayed using an electronic delay box (DB64, Stanford Research Systems) to ensure the timing of the synchronization signal matched the timing of the detected fluorescence. The FPGA used the synchronization signal from the laser to perform demultiplexing of the detected fluorescence and assign the signal to the appropriate imaging area. The fluorescence signal and the resonance synchronization signal for each of the four areas were then passed to the microscope software for display and storage. The microscope was controlled through a custom software, 'SCOPE', which controlled the stages, focus tunable lenses, half-wave plates' mounts, scanning, and image acquisition for the microscope (C++; http://rkscope.sourceforge.net/). The microscope software enabled independent scanning of each scan engine and allowed each resonance scanner to run at different frequencies. Control of the scanners and focus tunable lenses was achieved using two DAQ cards (PXI-6733, National Instruments), one for each scan engine. The half-wave plates' motorized rotation mounts were controlled by a USB controller (KDC101, Thorlabs). The stages for the sample were controlled via serial connection to a controller (FMC223, Steinmeyer Mechatronik) and the stages for the FPU's were controlled via local network connection to a controller (8SMC5-Ethernet, Standa).

**Point spread function measurements**. Fluorescent beads with a diameter of 0.5 μm (T7281, Invitrogen) were used to measure the point spread function of the microscope. Beads were embedded in a 1% agarose solution and then mounted on a slide with a well and a coverslip over the top of the gel. Imaging was performed through a coverslip and embedded in gel to mimic in vivo imaging conditions. To acquire images off axis up to a 4.8 mm FOV we moved the FPU to reach a distance of 1.5 mm in the sample and then tilted the 2D scanning mirror in order to reach 2.4 mm off axis in the sample. Image stacks of the beads were acquired at $0.13 \times 0.13 \times 0.4 \ \mu m^3$ voxel resolution. For analysis and display, image stacks were resampled to $0.4 \times 0.4 \times 0.4 \ \mu m^3$ voxel resolution in ImageJ. Each bead stack was resliced into XZ and YZ stacks and then a maximum intensity projection was performed. The maximum intensity projections for the XZ and YZ planes were then used to perform the FWHM measurements. A line was drawn through the center of the bead in the lateral and axial directions. Pixel intensities along that line were fit with a Gaussian and the full-width at half maximum values were determined based on the Gaussian fits.

For the simulated PSFs (Supplementary Fig. 6) at each ETL offset the RMS wavefront error was optimized by adjusting the depth of the focal plane in the sample. The Huygens illumination PSF was simulated at this depth and depths ±10 μm in 1 μm steps. The illumination PSF was saved at each plane and then squared to generate the excitation PSF[2]. The resulting PSF was analyzed using the same method as described above.

**Animal preparation and imaging**. All experimental procedures were approved by the Institutional Animal Care and Use Committee for the Charles River Campus at Boston University. In adult (8–14 week old) C57Bl6 mice, stereotaxic viral injections of AAV2/9.*hSynapsin1*.jYCaMP1s or AAV2/9.*hSynapsin1*.GCaMP7f were performed in L2/3 and L5 of the cortex, 300 and 500 μm below the pial surface (600 nL total volume, $6.8 \times 1011 \ gc/ml$). Four areas of the cortex were targeted for virus injection: S1 (AP −1.1 mm, ML 3.3 mm), S2 (AP −0.3 mm, ML 4.0 mm), and two areas in M1 (AP 1.1 mm, ML 0.6 mm; AP 1.1 mm, ML 1.2 mm). Optical access to the cortex was achieved by implantation of a "crystal skull", which is a curved cranial window designed to match the curvature of the mouse cranium and avoid compression of the brain[13] (LabMaker). The crystal skull was implanted over the left hemisphere. The right hemisphere was covered with dental cement so that a metal head post could be attached to allow head fixation. Animals were habituated to awake head fixation. Quadroscope imaging was carried out at 30 Hz frame rate (563 × 300 pixels per area) with the FPUs and imaging arms positioned to target each of the expression sites. An average of 75 mW of laser power was delivered per imaging site.

**Photo-damage assay**. Photo damage experiments were performed on two adult C57Bl/6 mice with viral expression of GCaMP7f activity and crystal skull implants. Thirty minutes of continuous frame scanning was performed for four excitation beams (75 mW per beam) positioned across S1, S2, and M1. Sixteen hours later, mice were transcardially perfused with 0.1 M PBS and 4% paraformaldehyde. Brains were postfixed in 4% paraformaldehyde overnight, then washed in 0.1 M PBS and cut into 50 μm coronal sections. Slices were incubated in a blocking solution of 10% normal goat serum and 1% Triton X-100, then washed three times in 0.1 M PBS. Alternating slices were labeled for two primary antibodies, either GFAP and Iba1 or HSP and Caspase-3, in 5% normal goat serum and 0.1% Triton X-100. The primary antibodies used were: mouse monoclonal anti-GFAP (G3893; Sigma-Aldrich; 1:1,000 dilution), rabbit anti-Iba1 (019-19741; Wako Chemicals; 1:500 dilution), rabbit anti-cleaved caspase-3 (Asp175) (9661; Cell Signaling; 1:250 dilution), and mouse anti-HSP70/HSP72 (C92F3A-5) (ADI-SPA-810-D; Enzo Life Sciences; 1:400 dilution). Slices were washed three times in 0.1 M PBS and incubated in secondary antibodies for four hours (1:500 dilutions). GFAP and HSP were labelled with goat anti-mouse Alexa Fluor 647 (Invitrogen, A21235), and Iba1 and caspase-3 were labelled with goat anti-rabbit Alexa Fluor 555 (Invitrogen, A21429). Slices were washed and mounted with Fluoromount-G mounting medium (0100-01, SouthernBiotech), then imaged with a Nikon ECLIPSE Ni-E microscope and NIS-Elements software (Nikon Instruments). Relative fluorescent intensity compared to the contralateral hemisphere was measured for eight areas of laser exposure (4 × 2 mice) and in ten control areas with no laser exposure (4 and 6 per mouse). For each area, mean fluorescent intensity on the treated side was divided by mean intensity on the contralateral side.

**Animal behavior**. Mice were housed 2–3 per cage in reverse 12 h light cycle conditions at 72 °F and 50% humidity. All handling and behavior occurred under simulated night time conditions. One week following chronic window implantation, mice were handled daily for 1 week while acclimated to a minimum of 15 min of head fixation. Mice were water restricted and trained to a go/no-go texture discrimination task previously described[15]. Imaging during behavior began once animals reached a performance level of d′ > 1.75 (80% correct) for one session. Quadroscope imaging was carried out at 30 Hz frame rate (563 × 300 pixels per area) with the FPUs and imaging arms positioned to target S1, S2, and M1. An average of 75 mW of laser power was delivered per imaging site.

**Whisker tracking and analysis**. High-speed videography of whisker movement was acquired at 500 Hz as previously described[22,23]. For analysis, whiskers were automatically traced[23]. The angle, was extracted for all traced whiskers. Using the mean whisker angle, a Hilbert transform was applied to determine whisking amplitude, phase, setpoint, and the reconstructed whisker angle[17]. The position of the rotor was automatically tracked in the video using custom scripts in MATLAB (Mathworks). Whisker-rotor touch was scored as events in which the tip of at least one whisker came into within <5 pixel radius of the rotor face with 'touch onset' defined as the first possible whisker contact at the beginning of the sample and test period and 'touch offset' defined as the last possible whisker contact at the end of the sample and test period. Time vectors of kinematic parameters were downsampled to the imaging frame rate for further analysis.

**Data analysis**. All image processing was performed in MATLAB (Mathworks). Two-photon images were first processed for motion correction using a piece-wise rigid motion correction algorithm[24]. Regions of interest corresponding to individual active neurons were manually identified and calcium time courses were calculated as $(F - F_0)/F_0$ where $F_0$ represents the bottom 8th percentile of fluorescence across a 10 s sliding window.

**Reporting summary**. Further information on research design is available in the Nature Research Reporting Summary linked to this article.

## Data availability
The image data from this study will be shared on an unrestricted basis and requests should be directed to the corresponding author.

## Code availability

The code used to control the microscope is hosted at https://sourceforge.net/projects/rkscope/ and access to the code can be obtained by emailing the corresponding author and requesting access. Analysis code can be found at https://github.com/common-chenlab/quadroscope.

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

## Acknowledgements

We thank G. Estrada for initial input on microscope design, C-H. Yu for assistance in microscope alignment, K. Kilic for guidance in animal surgeries, K. Maniar for assistance in analyzing data, N. Manjrekar and W. Wang for assistance in immunohistochemistry, K. Podgorski for viral reagents, and C-H. Yu and X. Ye for comments on the paper. This work was supported by grants from a NARSAD Young Investigator Grant from the Brain & Behavior Research Foundation, the Richard and Susan Smith Family Foundation, Elizabeth and Stuart Pratt Career Development Award, the Whitehall Foundation, National Science Foundation Neuronex Neurotechnology Hub (NEMONIC #1707287), National Institutes of Health BRAIN Initiative Award (R01NS109965), National Institutes of Health BRAIN Initiative Award (UF1NS107705), and National Institutes of Health New Innovator Award (DP2NS111134).

## Author contributions

J.l.C. initiated and supervised the study. M.C., I.A.C, S-W.P., J.N.S, and S.L.S designed and built the microscope. A.M.A performed animal surgeries and photo-damage experiments. M.C. and J.L.C. collected and analyzed the data. M.C. and J.L.C. prepared the paper.

## Competing interests

The authors declare no competing interests.
