## [Peer Review File · Nature Communications]

REVIEWER COMMENTS

Reviewer #1 (Remarks to the Author):

The manuscript "Flexible simultaneous mesoscale two-photon imaging of neuronal activity at high speeds" from the groups of Jerry Chen and Spencer Smith demonstrates an optical microscope, the Quadroscope, for large field 2P scanning microscopy reaching a $\varnothing 4.8\text{mm}$ field of view and simultaneous scanning of four brain regions. This is achieved by using a low repetition (31.25MHz) fiber laser which enables to split the excitation beam into four temporally delayed (8ns) beam paths. The 4 beams are then coupled with 2 scan engine arms enabling scanning with two independent set of imaging parameters. Each of the 4 beams can be laterally and axially positioned independently on the others by a 'focal plane unit' comprising an ETL and an x/y translation stage to control the axial and lateral position of the scan region within a z-range of 480 μm and a $\varnothing 4.8\text{mm}$ field of view. The system is coupled with two fiber lasers at 1040 and 920 nm, this enables 2P imaging of green as well as yellow and red genetically encoded Ca²⁺ indicators. The capabilities of the Quadroscope are demonstrated for in vivo imaging of neuronal population activity across 4 targeted area in mouse sensorimotor cortex at 30Hz. To prove the capability of the Quadroscope to track neuronal activity at behavioral relevant time scales, the authors performed simultaneous imaging of neuronal activity across primary somatosensory (S1) and motor (M1) cortex. Analysis of deconvolved calcium activity enabled to identify whisking-related responses.

Although the Quadroscope is an impressive demonstration of high-quality optical engineering, it represents only an incremental advance with respect to the very similar system, Diesel2p, demonstrated by the lab of S. Smith and published on bioRxiv in 2020 and with respect to an even older paper (Stirmann et al. 2016) published by the same group.

The manuscript on bioRxiv is only briefly mentioned as a reference (Ref. 22) for a custom objective while a clear description of the Diesel2p microscope and a comparison among the two microscopes is surprising missing. This is the more surprising considering that S. Smith is co-author of both manuscripts.

Many of the performances of the Quadroscope, here presented as novelties, have been already demonstrated with the Diesel2p set up, scan speed (up to 60Hz in the Diesel2p), the $\varnothing 4.8\text{mm}$ FOV and cellular resolution. Moreover, the use of a deformable mirror for adaptic optics, integrated in the Diesel2p scope, assures up to 1.5 fold smaller axial resolution, uniform in the whole FOV. The remaining novelty of the Quadroscope is in the use of (commercial) low repetition laser sources that enables splitting the excitation laser into 4 beams against the 2 demonstrated in the with the Diesel2p. As mentioned in my previous report, and agreed by the authors, the idea of temporal multiplexing has been already used by other groups before. Although in most cases this was limited to microscope with a fixed beam configuration, this is not the case of the Quadroscope, where temporal-multiplexing is coupled with two independent scan regions.

The authors have now added a new experiment (poorly explained: location of the scan regions, size, imaging rate??) showing simultaneous imaging in primary and motor cortex. In the rebuttal letter, the authors claim that this result is the real novelty of the manuscript. However, in my opinion, in the way the manuscript is written, the emphasis is still on a claimed novelty of the optical system which is not demonstrated. Moreover, it is not clear to me how this experiment can be exclusively performed with the Quadroscope and not equally doable with the Diesel2p.

In conclusions, despite the new data proving the possibility of performing imaging of green Ca²⁺ indicators and the new experimental demonstration of imaging from 2 scan area, I remain of the idea the manuscript lacks enough novelty for a publication in Nature communication.

Reviewer #2 (Remarks to the Author):

The authors have addressed all the technical comments of the reviewers, and more carefully characterized the optical limits of their system. The work represents an important advance of two-photon mesoscope imaging and indeed has capabilities that could find many uses in existing neuroscience experiments - one of which is demonstrated here in this manuscript on S1/M1 imaging during whisking. I'm skeptical the frame rates of this system are really adequate to properly measure phase tuning via calcium signals since mice whisking can occur at frequencies exceeding 20 Hz depending on context, but this is not really the point of the manuscript, although I might tone down that statement/analysis.

None of the advances themselves are extremely conceptually novel or dramatically expand the capabilities of existing mesoscopes, but rather this is a substantial advance since it outperforms existing microscopes in frame rate and multi-area sampling.

REVIEWER COMMENTS

Reviewer #1 (Remarks to the Author):

The manuscript “Flexible simultaneous mesoscale two-photon imaging of neuronal activity at high speeds” from the groups of Jerry Chen and Spencer Smith demonstrates an optical microscope, the Quadroscope, for large field 2P scanning microscopy reaching a Ø4.8mm field of view and simultaneous scanning of four brain regions. This is achieved by using a low repetition (31.25MHz) fiber laser which enables to split the excitation beam into four temporally delayed (8ns) beam paths. The 4 beams are then coupled with 2 scan engine arms enabling scanning with two independent set of imaging parameters. Each of the 4 beams can be laterally and axially positioned independently on the others by a ‘focal plane unit’ comprising an ETL and an x/y translation stage to control the axial and lateral position of the scan region within a z-range of 480 µm and a Ø4.8mm field of view. The system is coupled with two fiber lasers at 1040 and 920 nm, this enables 2P imaging of green as well as yellow and red genetically encoded Ca² indicators. The capabilities of the Quadroscope are demonstrated for in vivo imaging of neuronal population activity across 4 targeted area in mouse sensorimotor cortex at 30Hz. To prove the capability of the Quadroscope to track neuronal activity at behavioral relevant time scales, the authors performed simultaneous imaging of neuronal activity across primary somatosensory (S1) and motor (M1) cortex. Analysis of deconvolved calcium activity enabled to identify whisking-related responses.

Although the Quadroscope is an impressive demonstration of high-quality optical engineering, it represents only an incremental advance with respect to the very similar system, Diesel2p, demonstrated by the lab of S. Smith and published on bioRxiv in 2020 and with respect to an even older paper (Stirmann et al. 2016) published by the same group.

The manuscript on bioRxiv is only briefly mentioned as a reference (Ref. 22) for a custom objective while a clear description of the Diesel2p microscope and a comparison among the two microscopes is surprising missing. This is the more surprising considering that S. Smith is co-author of both manuscripts. Many of the performances of the Quadroscope, here presented as novelties, have been already demonstrated with the Diesel2p set up, scan speed (up to 60Hz in the Diesel2p), the Ø4.8mm FOV and cellular resolution. Moreover, the use of a deformable mirror for adaptic optics, integrated in the Diesel2p scope, assures up to 1.5 fold smaller axial resolution, uniform in the whole FOV.

The remaining novelty of the Quadroscope is in the use of (commercial) low repetition laser sources that enables splitting the excitation laser into 4 beams against the 2 demonstrated in the with the Diesel2p. As mentioned in my previous report, and agreed by the authors, the idea of temporal multiplexing has been already used by other groups before. Although in most cases this was limited to microscope with a fixed beam configuration, this is not the case of the Quadroscope, where temporal-multiplexing is coupled with two independent scan regions.

The authors have now added a new experiment (poorly explained: location of the scan regions, size, imaging rate??) showing simultaneous imaging in primary and motor cortex. In the rebuttal letter, the authors claim that this result is the real novelty of the manuscript. However, in my opinion, in the way the manuscript is written, the emphasis is still on a claimed novelty of the optical system which is not demonstrated. Moreover, it is not clear to me how this experiment can be exclusively performed with the Quadroscope and not equally doable with the Diesel2p.

In conclusions, despite the new data proving the possibility of performing imaging of green Ca²⁺ indicators and the new experimental demonstration of imaging from 2 scan area, I remain of the idea the manuscript lacks enough novelty for a publication in Nature communication.

It seems like the reviewer is not questioning the novelty of the Quadroscope with respect to other mesoscopes published earlier by Svoboda, Schnitzer, Culver, etc but only with specific respect to the very recent Diesel2P. Since the Quadroscope and Diesel2P were developed in tandem, it is to be expected that they share similar starting points, specifically the wide field of view optics and the dual scan engines. Beyond that, implementation and demonstration has gone in different directions. The integration of adaptive optics has been a main focus in the Diesel2P system, whereas Quadroscope development pursued additional optical engineering and new laser sources to scale up excitation sites. The consequence is Diesel2P has improved imaging resolution while the Quadroscope is able to image 4 (rather than 2) areas simultaneously. To pit these two specific systems against each other in terms of novelty is not our intention and might not be the appropriate criteria to evaluate the manuscript. In contrast to novelty, we think that there is significance in demonstrating a family of mesoscopic imaging systems unified by similar design concepts but extended for different functions and applications. As the reviewer eluded, the fact that Diesel2P could (in theory) be reconfigured into a Quadroscope with multiple excitation sites or for the Quadroscope to include adaptive optics speaks to the ease of customizability, and therefore, dissemination of these technologies. By analogy, one could argue the development of RCaMP was not novel because a GECI already existed in the form of GCaMP, but there is significance and impact to having a suite of different colored GECIs that can be applied for similar, yet distinct, applications. By that criteria, RCaMP stands to be acknowledged as a separate yet complementary technology to GCaMP. We believe the same logic applies when comparing the Diesel 2P and the Quadroscope.

We apologize if the above was not properly conveyed in the manuscript. To address this, we have added lines in discussion that compare the Quadroscope with Diesel2P and their synergistic utility (line 160).

Regarding the imaging conditions of the new animal experiments. The conditions are the same as for the prior animal experiments. This is restated in the methods: line 325.

Reviewer #2 (Remarks to the Author):

The authors have addressed all the technical comments of the reviewers, and more carefully characterized the optical limits of their system. The work represents an important advance of two-photon mesoscope imaging and indeed has capabilities that could find many uses in existing neuroscience experiments - one of which is demonstrated here in this manuscript on S1/M1 imaging during whisking. I'm skeptical the frame rates of this system are really adequate to properly measure phase tuning via calcium signals since mice whisking can occur at frequencies exceeding 20 Hz depending on context, but this is not really the point of the manuscript, although I might tone down that statement/analysis. None of the advances themselves are extremely conceptually novel or dramatically expand the capabilities of existing mesoscopes, but rather this is a substantial advance since it outperforms existing microscopes in frame rate and multi-area sampling.

Thank you for the feedback, we have toned down any strong claims of being able to resolve phase locked differences in whisking patterns (see line 152).